# Mechanochemical Activation of Waste Clay Brick Powder with Addition of Waste Glass Powder and Its Influence on Pozzolanic Reactivity

**DOI:** 10.3390/molecules29235740

**Published:** 2024-12-05

**Authors:** Csilla Őze, Nikolett Badacsonyi, Éva Makó

**Affiliations:** Department of Materials Engineering, University of Pannonia, P.O. Box 1158, H-8210 Veszprém, Hungary

**Keywords:** clay brick, glass, pozzolan, sustainability, mortar, CO_2_ emission, waste, mechanochemical activation, planetary ball mill

## Abstract

The availability of industrially used supplementary cementitious materials (SCMs, e.g., fly ash) decreases due to the rise in renewable energy sources and recycling technologies. Therefore, it is essential to find alternative SCMs (e.g., waste glass and clay brick powder) that are locally available. Accordingly, in this paper, the mechanochemical activation of clay brick waste (CBW) with abrasive glass powder (GP) and its pozzolanic reactivity are investigated. The mixtures of CBW and GP in mass ratios of 100:0, 75:25, 50:50, and 25:75 were mechanochemically activated for 15, 30, 45, and 60 min. The physical, chemical, and structural changes of the mixtures were examined by X-ray diffractometry, Fourier-transform infrared spectroscopy, scanning electron microscopy, and specific surface area measurements. The pozzolanic reactivity was characterized by the active silica content and the 28-day compressive strength of the binders (a mixture of ordinary Portland cement and activated material). The addition of GP favorably reduced the agglomeration and increased the active silica content of the activated mixtures (e.g., by 7–37% m/m at 15 min of mechanochemical activation). The 60 min of mechanochemical activation and the addition of 50% m/m of GP can increase the compressive strength by approximately 8%. Economically, the addition of 50% m/m of GP was found to be favorable, where only 30 min of mechanochemical activation resulted in a considerable increase in strength compared to that of the ordinary Portland cement.

## 1. Introduction

Fired clay bricks and glass are some of the most widely used building materials worldwide, and today, urbanization and urban rehabilitation construction/demolition activities generate significant amounts of clay brick waste (CBW) and glass waste (GW) [1]. Brick cutting and refining can also generate CBW powder, while GW typically occurs in municipal solid waste as a packaging material [1]. These materials are not biodegradable, leading to social and public health problems as they are disposed of in landfills [2].

Clay brick and glass waste have a high SiO_2_ and Al_2_O_3_ content, which makes them suitable for use as pozzolanic supplementary materials for cement [2,3,4]. Today, one of the main areas of research in the construction industry is the development of new supplementary cementitious materials (SCMs) that are accessible locally in significant quantities. The cement industry, in line with environmental directives, is aiming to promote a circular economy, green goals, and a sustainable future by developing new eco-efficient SCMs to reduce the energy demand and carbon dioxide emissions of the cement industry (currently responsible for 5–7% of global anthropogenic CO_2_ emissions) [5,6]. Nowadays, approximately 4 billion tons of cement are produced worldwide, resulting in ~3.5 million tons of carbon dioxide emissions [5]. Additionally, the application of CBW and GW as SCMs can contribute to reducing the volume of waste, delaying the depletion of raw materials for cement production, and solving the supply problems of currently used SCMs (e.g., fly ash and blast furnace slag) [7,8].

Previous studies [9,10,11,12,13,14,15,16,17,18,19,20,21,22,23,24,25,26,27,28,29,30,31,32,33,34] focused on the individual characteristics of waste glass powder (GP) and CBW as SCMs. The research related to GP and CBW materials has demonstrated that GP has a high SiO_2_ and a low Al_2_O_3_ content, whereas CBW has a lower SiO_2_ and a higher Al_2_O_3_ content. Studies on SCMs demonstrated that materials with high active SiO_2_ content when mixed with ordinary Portland cement (OPC) result in a binder with improved mechanical properties. Conversely, materials with high active Al_2_O_3_ increase the chemical resistance of mortar/concrete specimens [35,36,37].

Almost all previous studies of GP [9,10,11,12,13,14] have emphasized its ability to produce varying properties in hardened mortar/concrete dependent on the particle size. It was found that at larger particle sizes, the alkali-silica reaction (ASR) results in volumetric expansion, which can initiate cracking in concrete [9]. It was proved that GP with a particle size of less than 75 μm can reduce the risk of ASR and can be favorable for pozzolanic reactivity [12,13]. Furthermore, the combination of GP with more reactive pozzolans (e.g., silica fume and blast furnace slag) can lead to a reduction in the rate of ASR, as Ca(OH)_2_ interacts with the more reactive pozzolanic materials before the ASR [12,13]. GP particles have a lower water absorption capacity than cement particles, which results in improved workability and slump flow when substituting cement [13,14,32]. This property is especially beneficial in the production of self-compacting concrete. Generally, GP is a slowly reacting pozzolan; it can reduce or only slightly increase the compressive strength after 28 days, but it can induce a significant increase in strength after 90 days or longer curing periods [32].

In the case of CBW, some studies [15,16] examined the use of CBW as an aggregate in concrete, and it was found that CBW significantly reduced the workability and strength due to its high porosity. At the same time, CBW has amorphous SiO_2_ and Al_2_O_3_ content, which is important for pozzolanic reactivity, allowing its direct use as SCM [34]. After CBW was ground to below 150 µm, the produced clay brick powder (CBP) was already suitable as a substitute for cement [17]. Decreasing the particle size of CBP increased the specific surface area, which increased the uptake of Ca(OH)_2_ and the strength, as well as improved water permeability at the same level of cement replacement [18,19,20]. Some studies [19,23,24,25] reported an increase in compressive strength and water permeability at lower levels of cement replacement due to the favorable pozzolanic reactivity and the “filler effect”. The binders containing CBP exhibited greater sulphate resistance due to the formation of finer-structured pores, which also allowed a lower penetration of chloride ions [30,31]. Migunthanna et al. [33] studied a geopolymer containing fly ash, blast furnace slag, and CBP, and it was established that CBP could replace both blast furnace slag and fly ash. These studies have extensively investigated the use of CBW as SCM, but there is a lack of studies on the combined use of CBW and GP.

In recent years, mechanochemical activation has been increasingly considered by the scientific community and industry as an environmentally friendly solution to increase the reactivity of natural and waste materials compared to chemical and thermal activation [38,39,40,41,42,43,44,45,46,47]. In mechanochemical activation (MCA), materials are transformed physically and chemically by the action of mechanical energy [41,42,47,48]. MCA proved to be a favorable method for the production of SCMs because it is able to break the chemical bonds in the material, leading to the amorphization of the crystalline phases and an increase in reactivity. In the case of kaolin, it was observed that MCA breaks Al-O-Si and Si-O bonds [39,41,49,50]. As a consequence, easily available amorphous SiO_2_ and Al_2_O_3_ are formed, leading to an increase in pozzolanic reactivity. During MCA of kaolin, the particle size decreased for a given grinding time and then stopped decreasing or increased due to aggregation and agglomeration, causing a decrease in reactivity [41,48]. Therefore, it is important to find the optimal grinding parameters to achieve the most favorable reactivity and minimize the energy consumption of MCA [41,42,47,48]. Nowadays, MCA has been effectively applied to activate clays using a high-energy agitator bead mill, which can provide industrial-scale applications of MCA [51]. In addition, MCA could probably be used to increase the reactivity of waste materials such as CBW and GP, but no such study has yet been performed.

Several studies [9,10,11,12,13,14,15,16,17,18,19,20,21,22,23,24,25,26,27,28,29,30,31,32] have focused on the separate use of CBW and GP as SCMs, but to our knowledge, the combined use of CBW and GP and their mechanochemical activation have not been investigated. In our previous studies, we found that the mechanochemical amorphization of kaolinite and the pozzolanic reactivity can be significantly increased by the addition of abrasive pozzolanic particles (diatomaceous earth, silica fume, and trass) [41,47]. The aim of this study is to explore the effect of MCA of CBW on the pozzolanic reactivity and the role of abrasive GP particles in the activation of CBW and in the inhibition of secondary mechanochemical processes (such as agglomeration).

## 2. Results and Discussion

### 2.1. X-ray Diffraction (XRD) and Fourier-Transform Infrared Spectroscopic (FT-IR) Analyses and Determination of Active Silica Content

First, the effect of the addition of glass powder (GP) on the mechanochemical amorphization of the crystalline phases in CBW was investigated by XRD. The untreated CBW contains diopside (D), anorthite (An), and quartz (Q) as major, as well as muscovite (Mu), mullite (M), and hematite (H) as minor crystalline phases, and glassy phase as shown in the XRD pattern (Figure 1, CBW). The XRD pattern of the GP sample (Figure 1, GP) shows a low-intensity quartz (Q) peak and a strong broad peak in the range of 10–40° 2θ, which indicates the presence of a large amount of glassy and a small amount of crystalline quartz phase. In the XRD patterns of untreated mixtures (Figure 1, 75_CBW, 50_CBW, 25_CBW), the amount of crystalline phases and corresponding peak areas and intensities decreased with the increasing GP content. In the XRD patterns of the ground samples, the reflections of diopside (D), anorthite (An), dehydroxylated muscovite (Mu), mullite (M), and hematite (H) broaden and decrease in intensity with the increasing grinding time. This is due to the fact that high-energy grinding increases the density of the dislocations and/or decreases the crystallite size and enhances the amorphization of the crystalline phases [41,47,52,53]. Similar to previous research [41,47,49,54] in the field of mechanochemical activation, the reflections of the quartz (Q) phase were reduced less by grinding compared to the reflections of the other crystalline phases due to the high hardness of quartz. The XRD patterns of the samples with different GP contents show similar changes with the increasing grinding time.

In order to characterize more precisely the effect of GP on the mechanochemical activation of CBW, we determined the quantity of phases in the raw and activated samples using the Rietveld method. Figure 2 clearly illustrates that the quantity of amorphous/glassy phase increases with the GP content in the untreated samples. The addition of 25, 50, and 75% m/m of GP to CBW increased the glassy phase content of the mixtures from 40 to 55, 72, and 84% m/m, respectively. In general, during mechanochemical activation, the quantities of anorthite and diopside decreased significantly (by about half) with the increasing grinding time, while the amount of quartz decreased slightly (by about 1%). Among the minor crystalline phases, the amorphization of dehydroxylated muscovite was the most rapid (e.g., after 30 min for CBW), while mullite and hematite showed a slower rate of amorphization (e.g., still detectable in CBW at 60 min of grinding). The content of the CBW amorphous and glassy phases increased by 13, 18, 22, and 25% m/m after 15, 30, 45, and 60 min of mechanochemical activation, respectively. After 60 min of mechanochemical activation of CBW, the amount of anorthite decreased with 11% m/m, the amount of diopside with 9% m/m, and the amount of quartz with 1% m/m. After another 60 min of grinding, with the addition of 25% m/m of GP (75_CBW), 50% m/m of GP (50_CBW), and 75% m/m of GP (25_CBW), reduced the anorthite and diopside content by 7 and 7% m/m, 4 and 3% m/m, and 4 and 3% m/m, respectively. This means that increasing the GP content (glass content) does not accelerate the amorphization of the crystalline phases in CBW because the amorphization degree is higher without GP. Thus, the rate of mechanochemical amorphization is more determined by the grinding time (energy) than by the increase in GP content. The influence of microstress growth and the decrease in crystallite size lead to the broadening of reflections, complicating the precise determination of the amorphous and glassy phases. Overall, mechanochemical activation increased the amount of amorphous and glassy phases to a similar degree in the samples with different GP content. It can be expected that the amorphous SiO_2_ and Al_2_O_3_ produced by the amorphization of crystalline phases (diopside and anorthite) in CBW could increase the pozzolanic reactivity. To verify this, the active SiO_2_ content of the raw and activated samples was determined.

As the XRD results demonstrated that the major constituents of the ground samples are phases amorphous to XRD, FT-IR spectroscopy was applied to find relevant differences in their modified molecular structures. Since the obtained FT-IR spectra of the ground samples followed a similar pattern, the spectra of the mixtures ground for 0, 15, and 60 min are presented in Figure 3. The mechanochemically induced spectral changes were investigated in the 400–1200 cm^−1^ (SiO and AlO stretching and bending, as well as Al-O-Si bending) region. The raw GP sample (Figure 3, GP) has three broad characteristic bands in this region at around 1000, 770, and 450 cm^−1^, which can be attributed to the four-coordinated silica of soda-lime glass [55,56]. In the FT-IR spectra of the CBW sample (Figure 3, CBW), strong absorption bands were observed in the 850–1200 cm^−1^ region, which can be assigned to Si–O–Si and Si–O–(Al) stretching vibrations of SiO_4_ and AlO_4_ tetrahedral units [55,57]. Another strong region of absorption lies between 400–500 cm^−1^, which can be characterized as O–Si–O and O–Al–O bending vibrations of tetrahedral units in a random arrangement. Additional absorption bands with small intensity can be detected between 500 and 850 cm^−1^, which can be essentially determined as (Si, Al)–O–(Si, Al) symmetrical stretching vibration bands [55,57]. As shown in Figure 3, the addition of GP to CBW caused the disappearance of the individual vibration bands characteristic of CBW and the enhancement of the three strong vibration bands characteristic of GP. In the FT-IR spectra of the mechanochemically activated samples (Figure 3), with increasing grinding time and GP content, the characteristic stretching and bending vibration bands of the raw CBW were transformed into three broad bands at around 1000, 800, and 450 cm^−1^, indicating a radical structural change of the silicate and aluminosilicate phases of CBW and the formation of readily available silica and alumina groups.

Figure 4 shows the changes in active SiO_2_ content. The amount of active SiO_2_ content in CBW showed a slight increase with the grinding time (0, 15, 30, 45, and 60 min), aligning with the amorphization of the diopside and anorthite phases and an increase in amorphous content. Note that the addition of 25% m/m (75_CBW, 0 min), 50% m/m (50_CBW, 0 min), and 75% m/m (25_CBW, 0 min) of GP resulted in a decrease in the amount of active SiO_2_ as the glassy phase increased. This suggests that approximately 69% m/m of the SiO_2_ in GP was inactive. The active silica content of mixtures with GP increased with time of mechanochemical activation similarly to the CBW sample. However, the amount of active SiO_2_ increased significantly after only 15 min of mechanochemical activation by 7, 26, and 37% m/m with the addition of GP. The highest active SiO_2_ content (51% m/m) was obtained for the 25_CBW sample after 60 min of grinding. For this sample, the increase in amorphous content related to the mechanochemical transformation of the crystalline phases was the smallest (Figure 2, 93% m/m (25_CBW_60)–84% m/m (25_CBW) = 9% m/m). This can be explained by the fact that 75% m/m of the sample is GP containing 99% m/m of glassy phase (not completely disordered), which could be transformed from an inactive to an active form by the mechanochemical activation. It can be concluded that the GP addition and the mechanochemical activation favorably increase the active silica content of the CBW and the mixtures.

### 2.2. Scanning Electron Microscopic (SEM) Analysis and Determination of Specific Surface Area (SSA)

SEM analysis can be used to directly determine the morphological changes during mechanochemical activation and the primary (PPS) and secondary (SPS) (aggregate and agglomerate) particle sizes [41,47]. The evaluation of this is important because during mechanochemical activation, gradually decreasing efficiency is observed as the milling time increases. Research clearly distinguishes three main stages during mechanochemical treatment: the Rittinger stage, where the interaction of particles is negligible and the SSA is proportional to the grinding time; the aggregation stage, where the degree of dispersion is still increasing, but the new surface area generated is not in relation to the energy input due to the interaction of particles; and the agglomeration stage, where strong chemical bonds between particles and irreversible agglomeration cause a decrease in dispersion [46]. The formation of agglomerates negatively affects the reactivity of the mechanochemically activated samples. Changes in primary and secondary particle size, as well as SSA, closely correlate with this phenomenon [41,47,48]. Therefore, the objective of the determination of the PPS, SPS, and SSA values was to clarify the influence of GP on the aggregation and agglomeration of CBW and to identify the Rittinger, aggregation, and agglomeration stages of grinding [48,58]. Figure 5 shows the secondary (SPS) and primary (PPS) particle sizes and morphology of the raw CBW (1/A), 75_CBW (2/A), and GP (3/A), as well as activated CBW (1/B, 1/C) and 75_CBW (2/B and 2/C) samples at smaller and higher magnification. In the SEM images of CBW (1/A), the porous aggregates (1.5–31 μm in diameter) consist of primary particles of varied shapes. The SEM images of the 75_CBW sample (2/A) show mainly porous CBW aggregates and some GP particles with sharp and cracked morphology. In the SEM images of the GP sample, these sharp and cracked primary particles (0.3–50 μm) dominate, which are typical of glass powder.

The SEM image (1/B) of the CBW sample ground for 15 min shows an increase in the size of SPS and a decrease in the size of PPS, together with the formation of spherical particles. After 60 min of grinding (1/C), the SPS significantly increased, but the PPS did not change notably. Increasing the GP content led to a favorable decrease in SPS for samples that were activated for 15 min (2/B) and 60 min (2/C). The PPS particles in the mixtures were rounded off due to the mechanochemical activation, similar to CBW, but their size did not change significantly after 15 and 60 min of activation. SEM images of the samples ground for 30 and 45 min (not presented) at 30- and 45-minute grinding show similar changes of morphology and particle size.

Subsequently, the SEM images were analyzed in more detail to determine the secondary (SPS) and primary (PPS) particle-size distributions of the samples ground for different times. The cumulative distribution (D) curves are presented in Figure 6 and Figure 7. For clarity and ease of interpretation, the figures show only the D curves of the samples activated for 15 and 60 min. After 15 min of grinding (CBW_15), the SPS distribution curve (Figure 6) of the CBW sample has already significantly shifted to larger particle sizes, indicating the formation of aggregates. After 60 min of grinding (CBW_60), the SPS distribution curve is shifted toward even larger particle sizes due to increased aggregation during mechanochemical activation. The addition of 25, 50, and 75% m/m GP at 15 min of grinding (75_CBW_15, 50_CBW_15, 25_CBW_15) shifts the SPS curves toward progressively smaller particle sizes compared to those of CBW and GP. The SPS curve of the sample containing 25% m/m GP sample after 60 min of grinding (75_CBW_60) shows a significant shift to larger particle sizes (compared to 15 min). Meanwhile, the SPS curve of the sample containing 50% m/m GP sample (50_CBW_60) shows only a slight shift to larger particle sizes. At the same time, the SPS curve for the sample with 75% m/m GP content (75_CBW_60) was not changed markedly. This indicates that the aggregation decreases with the increasing GP content, meaning that the addition of GP can significantly reduce the aggregation of CBW. In Figure 7, the strong leftward shift of the PPS curves of the CBW sample indicates a reduction in PPS, which was similar for 15 and 60 min of grinding. The PPS curves of the activated samples show an increase in PPS as the GP content increases, shifting more to the right of the activated CBW curves. This is due to the fact that the increasingly abundant hard GP particles are more difficult to grind than CBW grains. The SPS and PPS distribution curves of samples activated for 30 and 45 min show a similar phenomenon.

To ensure comparability, the characteristic particle sizes (D_10_, D_50_, and D_90_) were determined for the samples using the cumulative distribution curves for all samples, as presented in Table 1. To identify the different stages of the grinding process, the specific surface area (SSA) values are also given in Table 1. The SSA values of the CBW sample increased significantly by approximately two times during 60 min of mechanochemical activation. At the same time, the characteristic D values for SPS of the CBW samples significantly increased by about five times. The D values for PPS of the CBW samples decreased to one third after only 15 min of grinding, which did not change significantly with grinding time. The characteristic D values for SPS of the mixture with 25, 50, and 75% m/m GP content decreased up to 30 and 45 min of grinding and then increased due to aggregation or agglomeration. At the same time, the D values for PPS increased with the addition of GP. It is important to note that the SSA values reported in Table 1 show that the addition of GP reduced the SSA values of the mixtures. However, the maximum SSA value of 5 m^2^/g was reached after only 15 min of grinding, independently of the GP content. This is 2.5–5 times the SSA value of unactivated samples. This may be connected to the decrease in PPS values, which decreased significantly after 15 min of grinding and then showed no change. Overall, considering the distribution curves, the characteristic particle sizes, and the SSA values, the Rittinger stage is completed, and the aggregation stage started after 15 min of grinding in all cases. However, the agglomeration stage, which would lead to a decrease in SSA values, did not occur until 60 min of grinding [48,58].

### 2.3. Analysis of Pozzolanic Reactivity with Strength Measurement

The influence of mechanochemical activation on the pozzolanic reactivity of SCM can also be characterized by measuring the compressive strength of the binder (cement and additive) [38,39]. Figure 8 shows the 28-day compressive strength of the binder produced from activated mixtures compared to those of ordinary Portland cement (OPC) as well as the binder containing raw GP and CBW. Replacing 10% m/m of OPC with raw GP and CBW, the compressive strength decreased by 10 MPa and 3 MPa, respectively. The compressive strength of the binder containing CBW activated for 15 min reached that of OPC. As the grinding time increased, the compressive strength of this binder increased (e.g., by 7% and 18% after 15 and 60 min of grinding, respectively, compared to the raw CBW). All binders containing activated mixtures show a similar increasing trend in the compressive strength with the increasing grinding time as CBW. Nonetheless, the addition of GP at the same grinding times leads to a reduction in the 28-day compressive strength, even with an increase in the active SiO_2_ content and a decrease in the aggregate size. This is explained in the literature [45] by the fact that pozzolans with higher Al_2_O_3_ content react faster with Ca(OH)_2_, while the SiO_2_-rich pozzolans react slower, so the increase in compressive strength at 28 days may be even smaller. It is noteworthy that all binders containing activated samples reached and exceeded the compressive strength of the OPC, considering the standard deviation.

### 2.4. Analysis of the Cost-Efficiency of Mechanochemical Activation

The characterization of the activation efficiency in mechanochemical treatments is important to minimize the grinding costs. The relative strength-difference index is defined as the percentage value of the difference of the compressive strength of the binder and OPC compared to the compressive strength of the OPC (the control sample) [59,60]. The relative strength-difference index is positive if the supplementary cementitious material increased the strength compared to the reference and negative if it decreased it. Figure 9 displays the correlation between the percentage change of the relative strength-difference index and the specific energy demand of mechanochemical activation, aimed at describing the activation efficiency. In Figure 9, the relative strength-difference index of all mixtures increases with the increasing grinding energy. The addition of GP shifts the efficiency curves to the right of the CBW curve, indicating the higher grinding energy needed to achieve a given relative strength-difference index. It is remarkable that a strength-difference index greater than 0 can be achieved with just 15 min of mechanochemical activation of CBW. An almost comparable relative strength-difference index can be achieved with the addition of 50% m/m GP with the lowest specific energy (30 min of mechanochemical activation) among the samples containing GP. The results may suggest that the addition of a GP is unnecessary, as it does not improve cost efficiency. In the context of CBW mechanochemical activation, it is important to note that aggregation resulted in the significant embedding of CBW particles within the grinding jar, a phenomenon that can be reduced with an increase in GP content. This is important for the applicability of grinding, the preservation of the mills, and economic reasons.

## 3. Materials and Methods

### 3.1. Materials and Mechanochemical Activation

In this study, we used the powder of clay brick waste (CBW) produced from the polishing and cutting processes of fired clay bricks (Leier Hungária Kft. (Devecser, Hungary)). The material was pre-ground with a disc gap of approximately 100 μm using a Fritsch Disc Mill Pulverisette 13 classic line. The glass waste powder exhibited a nominal particle size of less than 75 μm, and it was produced by Daniella Industrial Park Ltd. (Debrecen, Hungary) The clay brick waste (CBW) was mixed with 0, 25, 50, and 75% m/m of glass powder (GP) for mechanochemical activation (the symbols of the samples containing 100, 75, 50, and 25% m/m of CBW are CBW, 75_CBW, 50_CBW, and 25_CBW).

The chemical composition of CBW, GP, and CEM I 42.5 N ordinary Portland cement (OPC, Duna-Dráva Cement Ltd. (Vác, Hungary)) was analyzed with the fused sample preparation method using a Philips Axios PW 4400/24 wavelength dispersive X-ray fluorescence spectrometer. The chemical composition of the materials is presented in Table 2.

The mechanochemical activation was performed using a Fritsch Planetary Mill Pulverisette 5/4 classic line. Based on our previous work [47], the rotational speed was chosen as 330 rpm, and the mass ratio of sample to grinding balls was chosen as 1:14. During dry grinding, each sample was mechanochemically activated in a 500 cm^3^ steel grinding jar with 121 steel balls with a diameter of 10 mm. The grinding times were 15, 30, 45, and 60 min, which are indicated in the symbol of the samples. (For example, the symbol of the CBW sample ground for 60 min is CBW_60.) According to our previous research, the possible contamination of samples by grinding is approximately 350 µg/g of iron content [47].

### 3.2. Analysis Methods

X-ray diffraction (XRD) effectively characterizes the amorphization processes that occur during mechanochemical activation [41,42,47,58]. The mechanochemical amorphization of CBW with the addition of GP was examined using a Philips PW 3710 diffractometer. The diffractograms were measured with CuKα radiation at 50 kV and 40 mA. The changes in amorphous and glassy contents with the grinding time were examined by the Rietveld full-pattern-fitting method, where the samples were spiked with 10% m/m of zinc oxide (internal standard). The Rietveld method compares the complete measured XRD pattern with calculated diffractograms of all phases based on their crystal structure data [61,62]. The amorphous phase content can be determined using a suitable internal standard with a known amount. The quantification error was around 1%. Diffractograms were collected in the range of 4–40° 2θ for the qualitative analysis and in the range of 4–70° 2θ for the Rietveld analysis. The data collection used the X’Pert Data Collector program, and the analysis of the diffractograms was carried out with the ICDD PDF-2 2021 reference database using the HighScore Plus program.

Fourier-transform infrared (FT-IR) spectroscopic measurements were performed with a Bruker Vertex 70-type spectrometer using a Bruker Platinum ATR adapter without additional sample manipulation. Spectra were recorded with a room temperature DTGS detector at a resolution of 2 cm^−1^. In each case, 512 scans were averaged to increase the signal-to-noise ratio. The GRAMS/Al version 9.0 program (Thermo Fischer Scientific, Waltham, MA, USA) was used to manage and analyze the spectral data.

The specific surface area of the samples was determined by the Brunauer, Emmett, and Teller (BET) method [63] with nitrogen adsorption at −196 °C using a 3Flex Micromeritics instrument. The samples before the measurement were treated in a vacuum at 30 °C for 4 h.

The morphology of the raw and mechanochemically activated samples, as well as the distribution of secondary (SPS) and primary particle sizes (PPSs), was examined by a Thermo Fisher Apreo S scanning electron microscope (SEM). Images were taken in high-vacuum conditions with a secondary electron (SE) detector at magnifications of 250, 500, 1000, 2000, 5000, 25,000, and 50,000. The microscope during the measurements was operated at 2 kV accelerating voltage and 50 pA beam current. The values of SPS and PPS were determined using ImageJ 1.54k software. The SPS was determined from images taken at 250× and the PPS from images taken at 25,000× magnification, where three different areas of a given sample were imaged, and the average of the three distribution curves was plotted. Detailed specifications of the particle size distribution curves are given in Appendix A.

To characterize the pozzolanic reactivity, standard cement mortar specimens (MSZ EN 196-1, 2016) were produced, where 10% m/m of the OPC was replaced with activated mixtures [64]. The mortar test specimens were prepared from 405 g of ordinary Portland cement (OPC), 45 g of mechanochemically activated mixture, 225 g of water, and 1350 g of sand. Appendix A contains a detailed description of the compressive strength measurements.

The pozzolanic reactivity of the mixtures was also characterized by the active SiO_2_ content. The total SiO_2_ content and the inactive SiO_2_ content of the samples were determined (according to MSZ EN 196-2) [65]. The active SiO_2_ content of the samples was calculated by the difference of total and inactive SiO_2_. The standard deviation measurements were 0.22% m/m. Appendix A provides a detailed description of the determination.

## 4. Conclusions

This study investigated the effect of the addition of GP on the mechanochemical activation of CBW and the resulting changes in crystal structure, morphology, and pozzolanic reactivity. The following main conclusions can be derived from this study:The mechanochemical amorphization of the crystalline phases in CBW was mainly increased by the grinding time (energy) and not by the addition of GP. A total of 60 min of grinding, with the addition of 25, 50, and 75% m/m of GP (75_CBW, 50_CBW, and 25_CBW), reduced anorthite and diopside contents by approximately 7–3% m/m.The addition of GP together with the mechanochemical activation significantly increased the active silica content of mixtures of CBW (e.g., after 15 min of grinding, the amount of active SiO_2_ increased by 7, 26, and 37% m/m with the addition of 25, 50, and 75% m/m of GP), as the activation transformed the glassy phase of GP from an inactive to an active state.The increase in the GP content positively reduced the aggregation rate in the mixtures with CBW; thus, agglomeration of particles did not occur at longer grinding times.The 28-day compressive strength of the binder, made by the 15-minute mechanochemical activation of CBW, already reached that of the OPC. The compressive strength decreased slightly with the increasing GP content.Taking into account the relative strength-difference index and the specific energy demand, 50% m/m of the GP content and 30 min of grinding time were the most favorable for the mechanochemical activation of CBW to achieve a sufficiently high pozzolanic reactivity.

Overall, the reactivity of CBW was favorably enhanced by mechanochemical activation. From the results, it may seem that the addition of GP is unnecessary as it does not accelerate the amorphization and does not improve the compressive strength. However, it clearly reduced the aggregation and the adhesion of the ground material to the grinding jar, which is important from both a technological and economic point of view. Moreover, it is important to highlight that the use of GP can decrease the consumption of OPC and, consequently, the energy demand and CO_2_ emissions associated with the production process. Additionally, it contributes to the recycling of GP, which solves the burden on waste storage and mitigates air pollution.

## Figures and Tables

**Figure 1 molecules-29-05740-f001:**
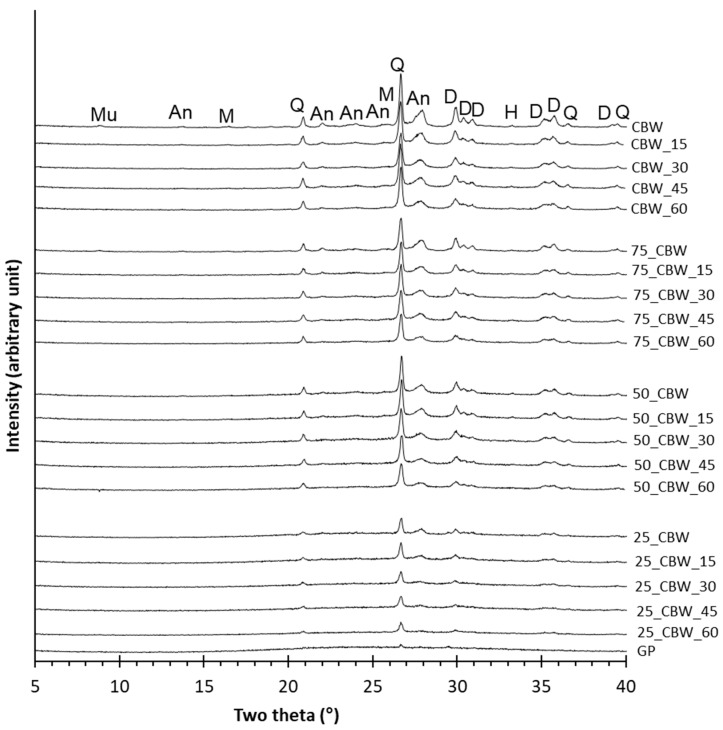
XRD patterns of the glass powder (GP) sample and the samples with 100, 75, 50, and 25% m/m of CBW (CBW, 75_CBW, 50_CBW, 25_CBW) ground for 0, 15, 30, and 60 min. (Mu: dehydroxylated muscovite PDF00-046-0741; An: anorthite PDF01-085-0878; Q: quartz PDF00-033-1161; D: diopside PDF00-011-0654, H: hematite PDF00-033-0664, M: Mullite PDF00-015-0776).

**Figure 2 molecules-29-05740-f002:**
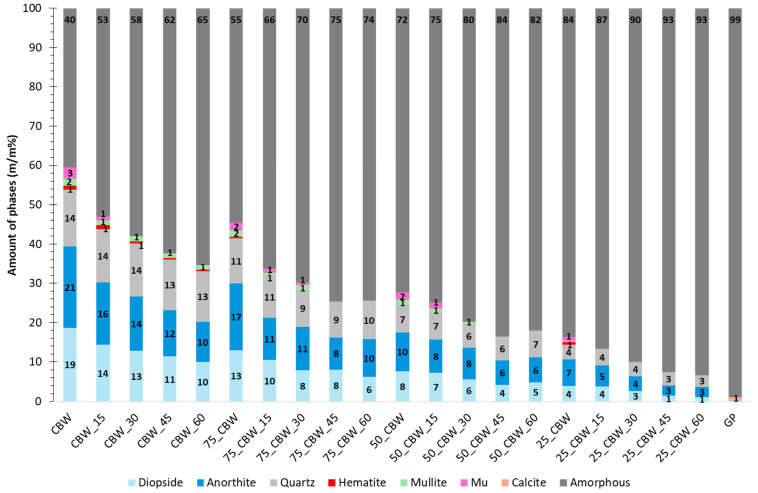
Phase composition of the samples ground for 0, 15, 30, and 60 min. (The symbols of samples are given in Figure 1).

**Figure 3 molecules-29-05740-f003:**
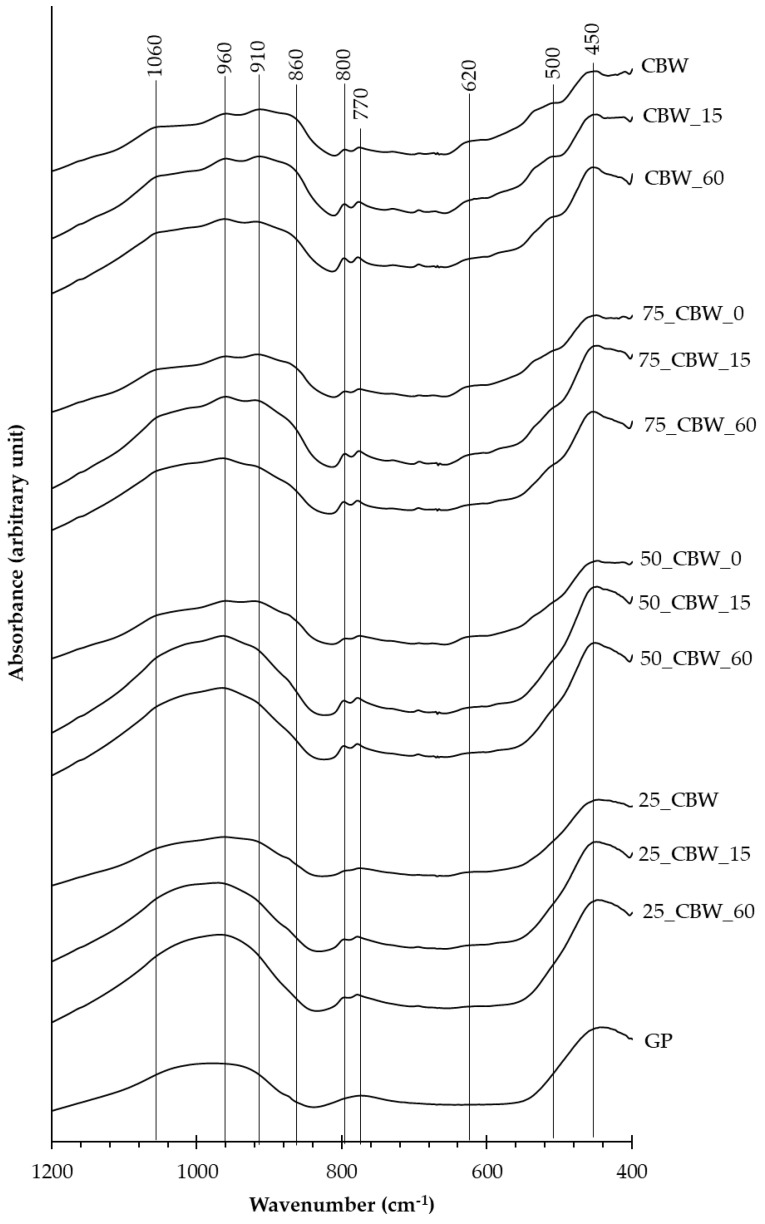
FT−IR spectra of the GP sample and the CBW, 75_CBW, 50_CBW, and 25_CBW samples ground for 0, 15, and 60 min. (The symbols of samples are given in Figure 1).

**Figure 4 molecules-29-05740-f004:**
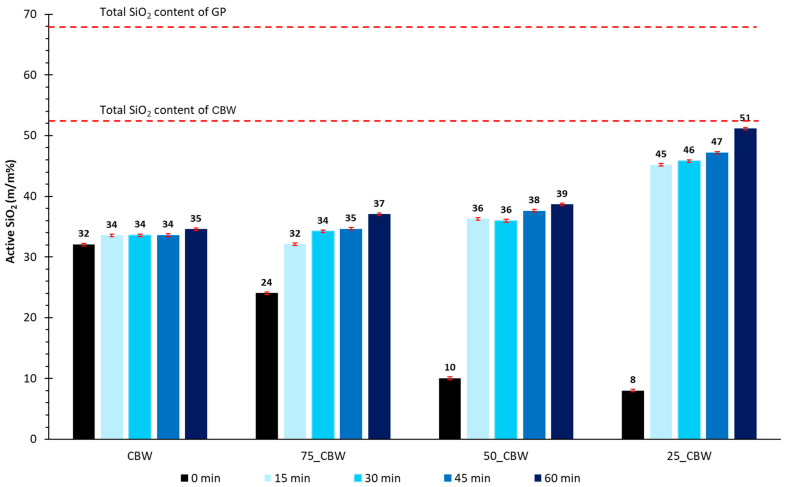
The amount of active SiO_2_ of the samples ground for 0, 15, 30, and 60 min. (The symbols of samples are given in Figure 1).

**Figure 5 molecules-29-05740-f005:**
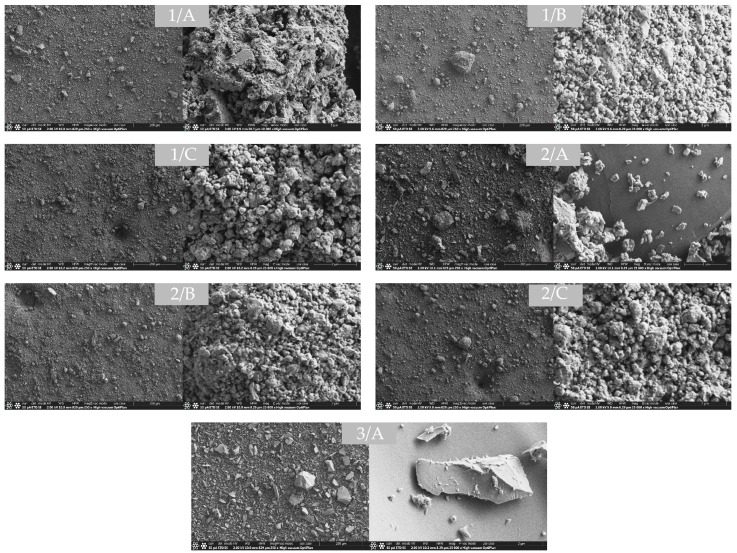
SEM images of the samples measured at 250× and 25,000× magnification: CBW (1/A), CBW_15 (1/B), CBW_60 (1/C), 75_CBW (2/A), 75_CBW_15 (2/B), 75_CBW_60 (2/C), and GP (3/A). (The symbols of samples are given in Figure 1).

**Figure 6 molecules-29-05740-f006:**
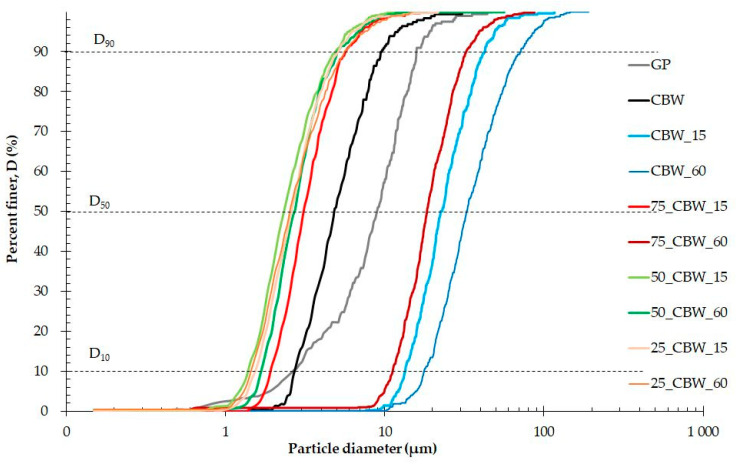
The secondary particle-size distribution (SPS) of the CBW, 75_CBW, 50_CBW, and 25_CBW samples after 15 and 60 min of mechanochemical activation and the primary particle-size distribution (PPS) of the GP sample. (The symbols of samples are given in Figure 1).

**Figure 7 molecules-29-05740-f007:**
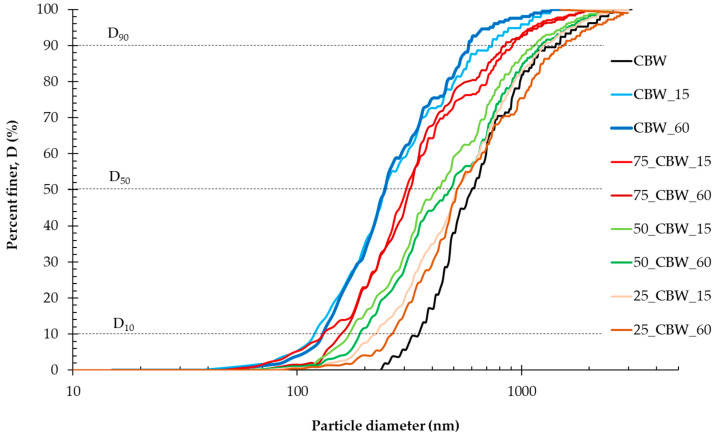
The primary particle-size distribution (PPS) of the CBW, 75_CBW, 50_CBW, and 25_CBW samples after 15 and 60 min of mechanochemical activation. (The symbols of samples are given in Figure 1).

**Figure 8 molecules-29-05740-f008:**
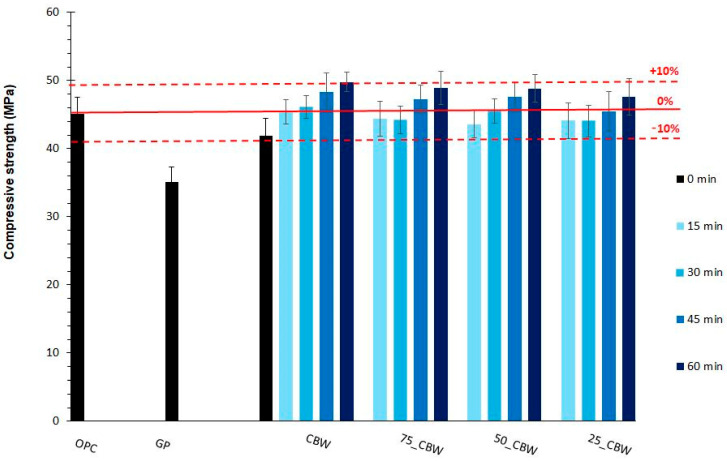
The compressive strength of mortar specimens prepared with the OPC, the GP, the CBW, the 75_CBW, the 50_CBW, and the 25_CBW samples after 15 and 60 min of mechanochemical activation. (The symbols of samples are given in Figure 1).

**Figure 9 molecules-29-05740-f009:**
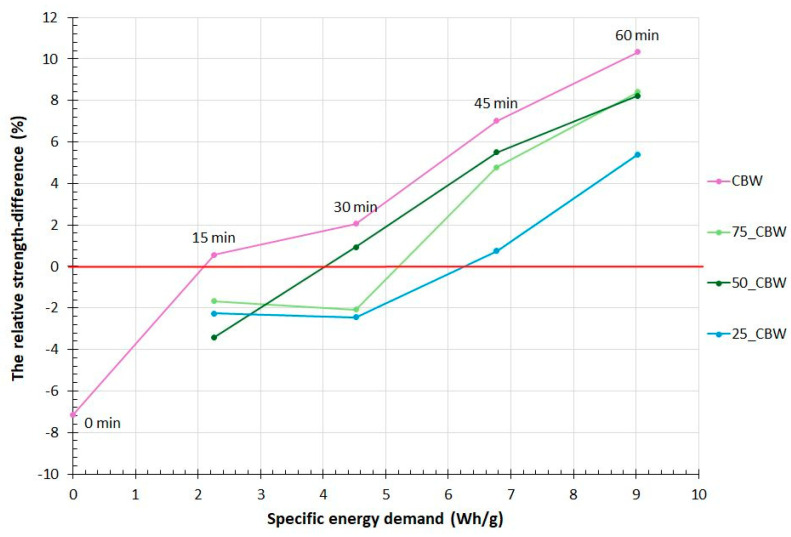
The relative strength-difference index as a function of specific energy demand of mechanochemical activation. (The symbols of samples are given in Figure 1).

**Table 1 molecules-29-05740-t001:** Particle-size characteristics of the samples determined from SEM images at different grinding times and their specific surface area (SSA). (The symbols of samples are given in Figure 1).

Symbol	SPS (μm)			PPS (nm)			SSA (m^2^/g)
	D_10_	D_50_	D_90_	D_10_	D_50_	D_90_	
GP	-	-	-	2650	8950	15,850	1
CBW	3.05	5.15	9.85	350	610	1420	3
CBW_15	13.55	22.50	42.50	130	240	750	6
CBW_30	12.25	23.90	49.60	125	250	790	6
CBW_45	15.10	25.70	46.10	145	215	710	6
CBW_60	17.60	32.50	71.15	130	245	580	5
75_CBW							2
75_CBW_15	2.00	3.20	5.75	125	310	910	5
75_CBW_30	1.60	1.70	5.10	120	330	800	5
75_CBW_45	1.80	3.35	7.65	145	310	865	5
75_CBW_60	11.25	18.50	32.45	160	320	850	5
50_CBW							2
50_CBW_15	1.55	2.60	5.10	180	430	1150	5
50_CBW_30	1.55	2.45	5.10	175	400	1200	5
50_CBW_45	1.60	2.65	5.70	185	460	1100	5
50_CBW_60	1.70	2.70	5.05	190	480	1230	5
25_CBW							1
25_CBW_15	1.40	2.35	4.90	225	515	1255	5
25_CBW_30	1.45	2.35	4.25	245	540	1195	5
25_CBW_45	1.45	2.15	4.10	220	530	1350	5
25_CBW_60	1.45	2.55	5.60	275	525	1525	5

PPS, primary particle size determined by SEM. SPS, secondary particle size determined by SEM. D_10_, D_50_, and D_90_ are the particle diameters at 10%, 50%, and 90% of the undersize (D) value, respectively, of the PPS and SPS distribution. The errors in the determination of SSA are ±1 m^2^/g.

**Table 2 molecules-29-05740-t002:** Chemical composition of CBW, GP, and OPC.

Component	CBW	GP	OPC
	(%m/m)		
SiO_2_	53.35	70.15	20.54
Al_2_O_3_	16.48	1.77	5.40
Fe_2_O_3_	6.20	0.45	3.26
TiO_2_	0.67	0.07	0.27
CaO	12.83	10.60	61.97
MgO	5.46	2.82	1.61
K_2_O	2.95	0.56	0.52
Na_2_O	0.68	13.42	0.11
SO_3_	0.28	0.04	3.16
Mn_2_O_3_	0.16	0.01	-
P_2_O_5_	0.12	0.02	-
SrO	0.05	0.06	-
ZnO	-	0.02	-
loss on ignition	0.74	-	2.74
Cl^−^	-	-	0.01

## Data Availability

The data presented in this study are available on request from the corresponding author.

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
