# Peer review of "Mechanochemical Activation of Waste Clay Brick Powder with Addition of Waste Glass Powder and Its Influence on Pozzolanic Reactivity"

_molecules, 2024, doi:10.3390/molecules29235740_

Round 1

Reviewer 1 Report

Comments and Suggestions for Authors

This paper addresses the use of waste materials such as clay brick and glass powder after mechanochemical activation as supplementary cementitious materials (SCMs). The characterization of materials is done by XRD, SEM, and compressive strength. Some minor comments are as follows:

1) More quantitative results can be added to the Abstract/Conclusion.

2) Emphasize more the gap in the literature and the novelty of combining clay brick waste (CBW) with glass powder (GP) for mechanochemical activation earlier in the introduction.

3) There is no Section 3! And better to mention the Section of Materials first, followed by Results.

4) Conclusion: Provide a stronger justification for the use of GP beyond reducing aggregation (e.g., its environmental or economic advantages).

5) What about the practical challenges of implementing mechanochemical activation in large-scale operations? How this can be addressed?

6) Refining data presentation and language can further improve the quality.

Author Response

Reviewers,

Herein we send the response to the comments on our paper „ Mechanochemical activation of waste clay brick powder with addition of waste glass powder and its influence on pozzolanic reactivity” by Cs. Őze, N. Badacsonyi, and É. Makó. We would like to thank the Editor and Referees for their helpful suggestions on the basis of which the manuscript has been modified. The responses to the revision can be found in the attached document. We indicated the changes made in the text with red.

Reviewer 2 Report

Comments and Suggestions for Authors

The paper is suitable for this journal, showing interesting results focusing on the mechanochemical activation of waste clay brick powder with addition of waste glass powder and its influence on pozzolanic reactivity. However, this original must be improved up to the quality standards of Molecules. My recommendation is 'Major revision'.

Authors can prepare a revised version taking into account my suggestions, comments and corrections, as follows in this report. First of all, it is recommended that the experimental details described in “4. Materials and Methods” must be presented before the results and discussion (Section “2. Results and discussion”): then, they will be presented as “2. Materials and Methods” and “3. Results and discussion”.

Abstract and Keywords are considered adequate. Abstract includes the main points of this investigation, provide results and the main conclusions.

1.  Introduction: this part of the paper is well structured, although the authors indicate previous studies in a long list of references: 9-34 (Page 2, Line 45) and 9-32 (Page 3, Line 95). They commented the main findings in the next paragraphs (after Line 52, Page 2).

Line 76: Correct “filler effect. It must be “filler effect”.

Concerning 'Mechanochemistry' and recent years (Lines 81...), it is suggested an addional reference in this section to improve the quality of thr original. It is an interesting paper by several researchers, presented as a review concerning nanoparticles and technology. This reference is as follows:

Balaz, P.; Achimovicova, M.; Balaz, M.; Billik, P.; Cherkezova-Zheleva, Z.; Criado, J.M.; Delogu, F.;  Dutkova, E.;Gaffet, E.; Gotor, F.J.; Kumar, R.; Mitov, I.; Rojac, T.; Senna, M.; Streletskii, A.; Wieczorek-Ciurowa, K. Hallmarks of mechanochemistry: from nanoparticles to technology, Chem. Soc. Rev. 2013, 42, 7571-7637, doi: 10.1039/c3cs35468g

Finally, the aim of this study is indicated (Page 3, Lines 100-102).

 2. Results and Discussion: this part of the submitted paper must be presented after the description of “Materials and methods” (section 3. in this original).

Comments to improve this section as appears in the submitted paper are suggested:

2.1 X-ray diffraction (XRD)… The muscovite (Mu) identified by XRD besides mullite, hematite, diopside, anorthite, etc., is a dehydroxylated phase according to the nature of the sample: a fired clay brick. The firing temperature of treatment in the raw clay produced high-temperature phases, such as diopside and anorthite. The muscovite is a dehydroxylated phase, (although depending of the kinetic conditions, muscovite could be partially dehydroxylated, check literature on this subject.

Then, the figure caption where these results are presented (Figure 1) must be modified indicating that “Mu” is dehydroxylated muscovite.

Page 5, Line 173: The difference taking into account the results presented in Figure 2 is correct: 93 (25_CBW_60) – 84 (25_CBW) = 9 %.

Page 5, Lines 173-175: The disorder is at long-distance (then, amorphous to X-ray diffraction) but there is an order at short distance. The last can be studied using a technique such as IR spectroscopy or better solid-state nuclear magnetic resonance spectroscopy.

Page 6, Figure 3: The error bars of the results must be included.

Page 6, 2.2 Scanning..., Line 191: Rittinger. The authors can include a short explanation about this. It could be of interest to the readers. In page 9, Line 263, the authors indicate "the Rittinger stage is completed".

Page 7, Line 209: check and revise the indication of samples studied by SEM as mentioned in the text and the images included in Figure 4. It seems that for 15 min the picture is 2/B and for 60 min is 2/C.

For a better presentation of Figure 4: Magnifications of SEM images must be included.

Page 8, Figure captions: Figure 5 must include “SPS” and Figure 6 “PPS” as mentioned along the text.

Page 10, Table 1: The errors (+/-) in the determination of SSA must be included. 

Page 11, 2.4 Analysis of... The relative strength index (%) is defined, but what about the results presented in Figure 8 with negative value? It must be explained.

It should be noted that in reference 55 (Alqedra et al.) this parameter (Table 8 of ref. 55) is positive.It is a ratio.

2 .Materials and Methods: this part of the submitted paper must be presented before the description of “Results and discussion” (section 2. in this original submitted paper).

Page 12, Lines 344-350: It should be indicated the possible contamination of the samples by grinding using the media and the grinding jar.

Page 14, Lines 363-365: The quantitative method to determine the percentages of crystalline phases identified by XRD must be described (in short), providing reference(s). The errors must be indicated. According to Figure 2, some phases are in low percentages (5-1 %). What about the determination of the content of amorphous phase? 

Page 13, Lines 366-368: It is recommended a reference of the “BET method”. It must be indicated if the samples were heated (temperature?) and outgassed (conditions) before analysis. The errors in the determination of SSA must be provided (and the values in Table 1, as mentioned above).

Lines 374-376: More details of the procedure using this software must be described, in particular the errors in these calculations, number of determinations, etc. (Figures 5 and 6).

Line 384: The determination was performed following a standard procedure. A reference must be included.

The compressive strength measurements are not described: preparation of samples, method and equipment. In general, the number of tested specimens must be provided.

4. Conclusions: they are sound. The main conclusions of this investigation have been achieved, being very interesting and the change inactive to active state of GP by mechanochemical treatment.

Correct "sate" in Line 397. It must be "state".

Finally, consequences of the use of GP are deduced. 

Page 14

6. Patents: Revise. It  must be deleted because the text is a guideline.

Supplementary Materias: it must be deleted.

Institutional Review Board Statement an Informed Consent Statement: they must be deleted. The texts are guidelines.

Page 15, Acknowledgments: this part must be completed.

Appendix A: Correct a minor mistake in Line 456: "assium". It must be "potassium".

Appendix B: it must be deleted.

References: This part of the submitted paper must be revised in deep taking into account the guidelines of this journal to present the references. Suggestions to improve this section are presented as follows:

Reference 4: the journal is not included. Pleae, include it to complete this reference.

Ref. 5 and ref. 6: they must be completed with journal name, volume, pages, etc,

Ref. 32: it must be completed.

Ref. 35: include the name of the journal (abbreviated).

Ref. 46: include the name of the journal (abbreviated). According the doi, it must be "Minerals" (in italics).

Ref. 51: it must be completed with data.

First of all, the references are presented including …

Author Response

(The authors gave the same response as above.)

Round 2

Reviewer 2 Report

Comments and Suggestions for Authors

The authors have performed a good job. They revised the original submitted paper properly according to my report. Clarifications have been provided taking into account my suggestions and corrections. Answers to several questions are considered adequate.

Furthermore, a FTIR spectroscopy study has been included in the revised version. This study is interesting to complete the results of this investigation.

The original has been sufficiently improved. It has resulted in a clearer and much more compelling paper. Consequently, the paper can be accepted in the present form.

The section "5. Patents" could be deleted.